# A Comparative Study of Perceptions of Destination Image Based on Content Mining: Fengjing Ancient Town and Zhaojialou Ancient Town as Examples

Jiahui Ding [1] , Zheng Tao [1] , Mingming Hou [2], Dan Chen [1] and Ling Wang [1,*]

1   School of Design, Shanghai Jiao Tong University, Shanghai 200240, China; dingjiahui@sjtu.edu.cn (J.D.);
    zheng.tao@sjtu.edu.cn (Z.T.); danchen.gator@sjtu.edu.cn (D.C.)
2   Department of Development and Planning, Shanghai Municipal Agricultural and Rural Commission,
    100 Dagu Road, Shanghai 200003, China; mmhou@nyncw.shanghai.gov.cn
*   Correspondence: wwlling@sjtu.edu.cn

**Abstract:** Ancient canal towns in Jiangnan have become important tourist destinations due to their unique water town scenery and historical value. Creating a unique tourist image boosts these ancient towns' competitive edge in tourism and contributes significantly to their preservation and growth. The vast amount of data from social media has become an essential source for uncovering tourism perceptions. This study takes two ancient towns in Shanghai, Zhaojialou and Fengjing, as case study areas. In order to explore and compare the destination images of the towns, in the perception of tourists and in official publicity, machine learning approaches like word embedding and K-means clustering are adopted to process the comments on Sina Weibo and publicity articles, and statistical analysis and correspondence analysis are used for comparative study. The results reveal the following: (1) Using k-means clustering, destination perceptions were categorized into 16 groups spanning three dimensions, "space, activity, and sentiment", with the most keywords in "activity" and the fewest in "sentiment". (2) The perception of tourists often differs significantly from the official promotional materials. Official promotions place a strong emphasis on shaping the image of ancient towns based on their historical resources, presenting a more general picture. Tourist perception, which is fragmented, highlights emerging elements and the experiential activities, along with the corresponding emotional experiences. (3) Comparing the two towns, Fengjing Ancient Town stands out, with more diverse tourist perceptions and richer emotional experiences. This underscores the effectiveness of tourism activities that use space as a media to evoke emotions, surpassing the impact of the spaces themselves.

**Keywords:** ancient town; destination image; UGC; DMO; machine learning

## 1. Introduction

In the 1980s, the emergence of tourism in Zhouzhuang marked the genesis of ancient town tourism in the country [1]. For all ancient towns, tourism development presents a vital opportunity to strike a balance between preserving historical culture and fostering socio-economic growth. Nevertheless, due to the shared historical backgrounds in the Jiangnan region, issues such as homogenization, superficiality, and theatricality have surfaced within the development models of ancient towns. This has led to a noticeable uniformity in the imagery of various Jiangnan ancient towns. However, these issues seem to be overly generalized. Each ancient town merits a more profound examination, as they may not be entirely similar to one another. Furthermore, it is essential to consider how different stakeholders perceive these towns. Research on destination images has emerged as an approach to delve into more nuanced and comprehensive depictions, facilitating a deeper understanding of ancient towns. This, in turn, paves the way for tailored develop-

ment strategies for ancient towns to enhance their allure and competitiveness within the tourism industry.

A destination's image is directly correlated with marketing, branding, and tourists' willingness to choose a destination [2,3]. Research on destination image is instrumental in understanding tourist preferences, leading to a continuous enhancement of tourism product quality. This approach enables a destination to stand out from competitors in the competitive landscape, elevating its ecological niche within the market [4,5]. Destination images are often derived from two primary sources. One of these sources is the destination marketing organization (DMO), which utilizes marketing and advertising strategies to craft the image of the destination [6]. The second source is tourists, who convey their perceptions of the tourism destination through platforms such as social media, travelogues, or by communicating with others, resulting in a destination image based on user-generated content (UGC). While there are differences in how DMOs and UGC shape the destination image [7], both significantly impact the choices of tourists of destinations and decisions regarding travel behaviors [8]. The rapid advancement of information and communication technologies has shifted the focus of the tourism industry from traditional markets towards a digitally driven marketplace [3]. In addition to traditional printed materials, DMOs have started utilizing the internet as a means of promotion. Tourists increasingly share information and their tourism experiences on public platforms. UGC primarily originates from platforms such as Facebook, Instagram, Twitter, and Flickr [9], as well as platforms in mainland China like Sina Weibo, tourism websites, and Red. With the aid of social media platforms, both UGC- and DMO-crafted destination images have achieved broader dissemination, enhancing their impact on potential tourists.

The higher the overlap between the destination image shaped by the DMO and the image perceived by tourists, the more advantageous it is for the development of the destination [10]. This is because tourists expect DMO promotions to align with authentic travel experiences and are not receptive to "fake events", where the DMO promotes the destination image with purpose but not in a truthful manner [3,11]. While the significance of maintaining a consistent destination image between DMO and UGC has been emphasized, it is noteworthy that existing research on destination images often concentrates on individual stakeholders, rather than facilitating a comparative analysis that juxtaposes DMO and UGC [12–15]. Furthermore, research that systematically examines the disparities between DMO and UGC in both destinations remains scarce. To close the research gaps aforementioned, this study takes Fengjing and Zhaojialou ancient towns in Shanghai as case study areas and employs a content analysis of web text to address two research questions: (1) What is the level of alignment between tourists' perceptions of the ancient town destination image and the official promotional image? (2) Regarding different ancient towns, does the alignment between UGC and DMO destination images exhibit similarities? This study contributes to methodological advancements by employing machine learning techniques. By outlining textual content analysis, this research contributes to helping tourism marketers reflect on the balance between DMO and UGC and optimize the storytelling within their DMO content.

## 2. Literature Review

John Hunt, Edward Mayo, and Clare Gunn initiated research on destination image in the 1970s [16]. Early research methods for the perception of destination image, such as questionnaires, lacked reliable theoretical foundations, and the accuracy of data collection was often questioned. Additionally, comparability between different scales or questionnaires was lacking [17,18]. In the 1990s and beyond, more complex quantitative analysis methods emerged by means of the advanced state of destination image theory [17]. However, the strong preference for quantitative analysis led to a trend towards structured research, which in turn resulted in the overlooking of the unique characteristics of tourism destinations, and there was an incline in research perspectives—out of the 142 publications on destination image from 1972 to 2000, approximately 80% of the research focused on tourists, while studies

involving other stakeholders of destinations, such as DMO and local residents, accounted for only 2% of the research [16]. Traditional content analysis methods in destination image research have relied on sources such as interviews and questionnaires with tourists, literary works, and audiovisual materials featuring destinations. These sources often have a limited amount of data, which makes it challenging to explore the evolution of destination images over longer time spans. As global tourism and the internet have developed, destination image research since the year 2000 has not only increased its focus on other stakeholders but has also transformed data source selection. In the era of the internet, online data content offers a rich and abundant source of information, covering longer periods. This has become a crucial aspect of contemporary destination image research. Internet data, particularly online text, has become a prevalent and valuable data source for studying destination images due to its accessibility and scale [10,19–21].

Zhang and Zhang (2009) extracted the top 60 high-frequency words from online reviews of the Chenshan Botanic Garden in Shanghai, and they found that the most frequently mentioned element across 16 categories was the sentimental experience [22]. When there is a need to process larger volumes of data, machine learning emerges as a powerful strategy for effectively handling and interpreting substantial datasets [3,23], potentially leading to the discovery of more intriguing results. Koblet and Purves (2020) collected over 7000 tourist reviews of England's Lake District National Park and processed the reviews using machine learning methods, and they found that many fresh experiences reported by users cannot be found in the underlying data [24]. Song et al. (2021) employed machine learning approaches to analyze over 20,000 tourist reviews of the Las Vegas Strip, which revealed that, despite the Las Vegas Strip boasting numerous world-renowned hotels, casinos, and resort properties that symbolize the area, most visitors expressed dissatisfaction with staying exclusively in their own hotel. Instead, they preferred to explore various locations on foot to experience a variety of attractions [25]. This result highlights a discrepancy between the conventional image of a destination and how tourists perceive it. Költringer and Dickinger (2015) utilized a web content mining approach to analyze three different sources of data, including UGC, DMO, and media, and this study concluded that UGC contains the most diverse information compared with DMO and media, and the content of UGC does not conform to the image DMO conveyed [10].

Research methods for studying the destination image of ancient towns primarily encompass traditional approaches such as interviews, questionnaires, visitor-employed photography (VEP), and textual analysis of tourist comments. The analyzed data volume remains limited, and the involvement of machine learning has not happened yet. Zhang et al. (2019) utilized a combination of questionnaires and photo-elicitation interviews to analyze the different perceptions of destination image in Tongli from tourists and residents, and their results showed that tourists' perceptions of architecture, streets, and lanes were higher than local residents', while the perception of historical and cultural elements was lower [26]. Tan et al. (2018) conducted a text classification and frequency analysis on Weibo data and concluded that tourists' perceptions of Zhaojialou were primarily focused on food [27]. Xu et al. (2017) analyzed photos taken by tourists and found that tourists exhibited a stronger perception of ancient bridges and residential buildings, which validated the effectiveness of the promotional efforts of DMO [28]. Dong and Xu (2017) compared tourist photos from Mafengwo with official pictures of Fengjing ancient town. They concluded that tourist photos placed more emphasis on activities and details, showing minimal influence from official imagery [29]. They argued that this finding contradicted Urry's "hermeneutic circle". In fact, Urry and Larsen (2011) acknowledged that the hermeneutic circle is an oversimplified explanation, and that tourist photos possess the capability to shape a new destination image, rather than replicating photos promoted by DMO [30]. Previous studies of Fengjing and Zhaojialou ancient towns lacked the involvement of machine learning, leading to limitations in data volume. While some research has compared the UGC and DMO of Fengjing, their analysis was primarily based on photographs, which often exclude experiential aspects beyond visual elements [30]. This may impose certain limitations. In

contrast, the analysis of textual content could yield a different result. Therefore, this study takes a different approach by analyzing online comments and integrating machine learning. It aims to explore the perception of destination image in both ancient towns and investigate the discrepancies between tourist perceptions and DMO representations.

## 3. Methods

### 3.1. Case Study Area

In this study, Fengjing Ancient Town and Zhaojialou Ancient Town in Shanghai were selected as the case study areas (Figure 1). Although the two towns are located in the same city, Shanghai, there are differences in their geographic locations, historical and cultural backgrounds, and patterns of development. Fengjing Ancient Town retains the typical style of a water town, and is located in the southwest corner of Shanghai and the northwest corner of Jinshan District, which is the junction of Shanghai and Zhejiang; it is characterized by the regional culture of "Wu gen Yue Jiao" and was listed in the first batch of "China's historic and cultural villages" and the first batch of historic and cultural villages of Shanghai in 2005 [31]. Zhaojialou is located in Gexin Village of Pujiang Town, Minhang District, Shanghai, which is the origin of the cultivation culture in Shanghai. Zhaojialou retains many buildings from the Ming and Qing Dynasties and was recognized as one of "China's historic and cultural villages" in 2014.

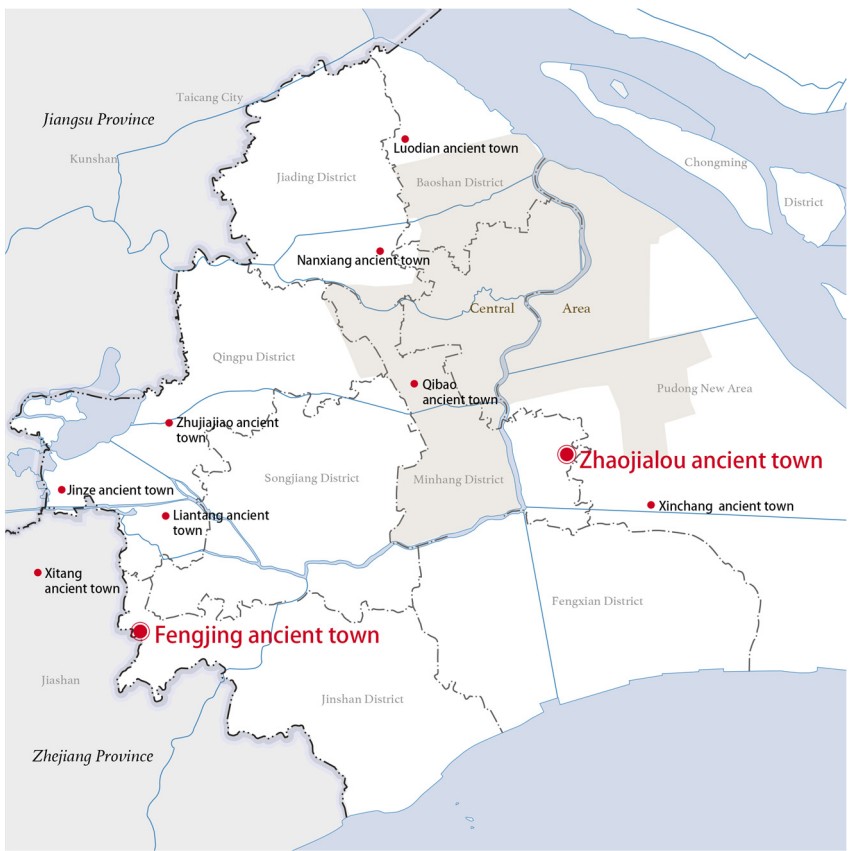

**Figure 1.** Geographical location of Fengjing and Zhaojialou (source: authors).

This study comprises four sources of a dataset encompassing UGC and DMO data from Fengjing Ancient Town (FJ) and Zhaojialou Ancient Town (ZJL). Regarding UGC data, the study employed keyword searches for the two towns to retrieve Sina Weibo hashtag check-in comment data spanning from 2010 to 2020. For Fengjing ancient town, the original textual data (FJ-UGC) consisted of 10,619 comments, while Zhaojialou ancient town (ZJL-UGC) had 11,485. The study primarily employed a content analysis of the

comments to establish tourists' perceptions of the destination image. Due to the absence of an official website for Zhaojialou, the study employed the Shanghai Cultural Tourism Promotion Website and three widely used tourism websites (Ctrip, Qunar, and Tuniu) to source official introduction articles for both ancient towns as DMO data. The DMO data for Fengjing (FJ-DMO) comprised five articles, and Zhaojialou (ZJL-DMO) consisted of four articles. All of the text materials consisted of Mandarin Chinese characters.

### 3.2. Data Processing

Text data were processed according to Figure 2.

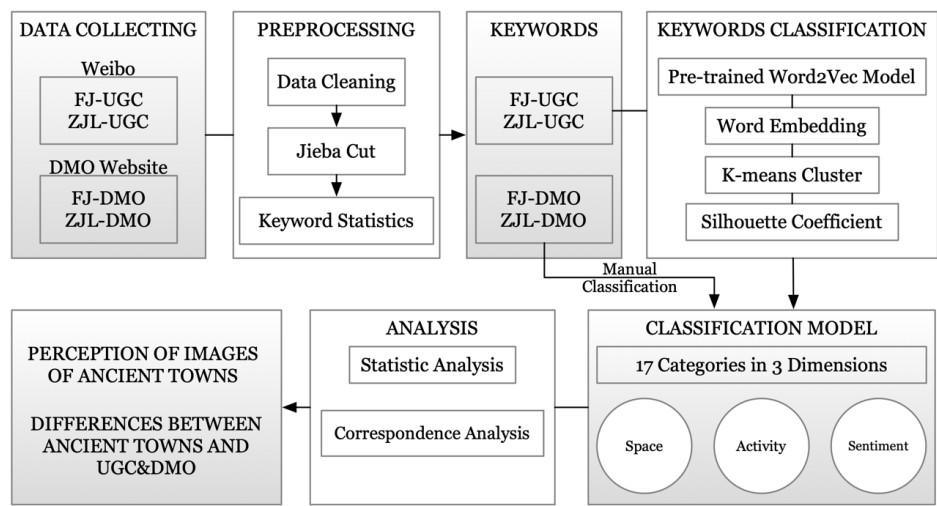

**Figure 2.** Route of data processing.

### 3.2.1. High-Frequency Keyword Extraction

The most frequently mentioned keywords in the comments, known as high-frequency keywords, represented the main points of tourists' attention and reflected their attitudes towards the destination [20]. Extracting high-frequency keywords involves three steps: data cleaning, tokenization, and statistics.

Blank comments and irrelevant content such as items only containing phrases like "share pictures", "I am at" in UGC data were removed. Additionally, duplicate comments were eliminated. As a result, the remaining comments of FJ-UGC were 9333, and ZJL-UGC had 10,716. Using regular expressions, only pure textual content without URLs and punctuation remained in both UGC and DMO data.

Jieba is a widely used Chinese text processing toolkit that is capable of segmenting text into words [23]. In this study, a user dictionary relevant to the ancient towns and a stop word dictionary including common pronouns, conjunctions, and unnecessary internet jargon were established. The UGC and DMO data, which had undergone data cleaning, were then subjected to tokenization using Jieba (github.com/fxsjy/jieba(accessed on 15 January 2022)). Word segmentation was performed multiple times due to Chinese words being composed of individual characters, which is different from English words. Many words identified during the initial segmentation, which needed to be excluded, were not present in the initial stop word dictionary. Additionally, some words are valuable and should be considered for inclusion in the user dictionary. The final segmentation results were established after refining the stop word dictionary and user dictionary.

Considering the disparity in data size between UGC and the DMO, distinct frequency thresholds were applied: keywords with a frequency greater than or equal to 5 for UGC and those with a frequency greater than or equal to 2 for DMO were selected for further analysis. Despite UGC data containing over 300,000 characters, significantly more than the 15,000 characters in DMO content, it is noteworthy that UGC offers limited valuable information, resulting in the exclusion of the majority of its content. DMO data are rela-

tively stable due to the articles being published by different institutions almost conveying the same content, whereas UGC exhibits substantial fluctuations that require more data to obtain more representative results. Moreover, it is worth noting that the presence of introductions from multiple institutions for both towns is rather scarce. The study has diligently sought to compile a comprehensive sample of articles released by DMOs. Additionally, the analysis primarily focuses on comparing percentages rather than absolute values, as this approach provides a more comparable perspective. Table 1 presents the count of the keywords finally selected and the corresponding frequency from the four data sources.

**Table 1.** High frequency keywords.

| Source | FJ-UGC | ZJL-UGC | FJ-DMO | ZJL-DMO |
|---|---|---|---|---|
| Number of keywords | 641 | 714 | 167 | 141 |
| Frequency | 9555 | 13,819 | 524 | 748 |

3.2.2. Keyword Classification

Content analysis is not simply a word frequency count but a reasonable way of classifying the data [32]. This study harnesses clustering algorithms to categorize keywords. On the one hand, the substantial volume of UGC data lends itself well to the application of machine learning in clustering analysis. On the other hand, employing a unified classification facilitates a horizontal comparison between UGC and DMO data. Consequently, the study exclusively performed clustering on the UGC data, which was subsequently applied to the DMO data as well.

(1)  Word embedding

To let computers "understand" human language, it is necessary to subject the keywords to word embedding processing, thereby converting them into word vectors [33]. The Word2Vec model is one of the most commonly used word embedding techniques [34,35]. After inserting the keywords into the Word2Vec model, corresponding word vectors were obtained. Following this, a clustering algorithm was applied to process these word vectors, effectively categorizing the unstructured textual data [20]. In this study, a pre-trained Word2Vec model released by Tencent AI Lab [36] was utilized to embed keywords and acquire their respective word vectors. Some of the keywords could not be embedded in the model due to the absence of corresponding vectors. These words were manually categorized after the clustering process.

(2)  K-means clustering

The study used K-means clustering to cluster the word vectors of FJ-UGC and ZJL-UGC. The silhouette coefficient was utilized to evaluate the clustering effectiveness and determine the optimal number of clusters in the K-means clustering [37]. The research discovered that the ideal k values for FJ-UGC were 21 and 24, while for ZJL-UGC, the optimal k values were 8 and 15 (Table 2). To ensure a consistent classification standard for the keywords of both ancient towns, the study initially chose the closest numbers, 21 and 15, as the initial cluster numbers for the keywords of Fengjing and Zhaojialou. This number was later unified based on human judgment.

(3)  Categories of keywords

The unification of the UGC keyword classification for both datasets was carried out through manual judgment, guided by the principle that "each category contains only a set of keywords with the same meaning or connotation" [38] and that "each category is distinct" [39]. Adjectives related to emotions were categorized into positive, neutral, and negative sentiments based on the Gooseeker platform's (gooseeker.com (accessed on 13 January 2022)) sentiment corpus. Based on this, the clustering results of the UGC data for both ancient towns were unified. Categories and keywords without actual significance

were omitted. This led to the development of a comprehensive classification method that includes sixteen categories under three dimensions (Table 3).

**Table 2.** Silhouette score of k-means cluster.

| K-means model of FJ-UGC | | | | | | |
|---|---|---|---|---|---|---|
| Cluster number | 10 | 15 | 20 | 21 | 24 | 25 | 30 |
| Silhouette score | 0.06210957 | 0.06513383 | 0.06720589 | 0.072402 | 0.072835 | 0.06117635 | 0.06356114 |
| **K-means model of ZJL-UGC** | | | | | | |
| Cluster number | 5 | 8 | 10 | 15 | 20 | 25 | 30 |
| Silhouette score | 0.0690023 | 0.075108 | 0.05928351 | 0.075067 | 0.05729202 | 0.06224708 | 0.06160499 |

**Table 3.** Categories with descriptions.

| Dimension | Category | Description |
|---|---|---|
| Space | Architectural Space | Enclosed spaces for various activities, such as shumai shops, restaurants, bars, etc. |
| | Gathering Space | Open spaces like markets, squares, ticket booths, and parking lots, where people gather or stay |
| | Street Space | Old streets, streets, alleys, lanes, streets, etc. |
| | Waterfront Space | Areas near rivers or formed by rivers, such as riverbanks, rivers, and both sides of the river |
| | Landmark Space | Clearly defined landmarks or buildings, like Zhihe Bridge, Heping Street, Wuyou Xian |
| Activity | Time | Specific time periods like months, days of the week, times of day, holidays, weekends |
| | External traffic | Modes of transportation such as buses, subways, driving |
| | Individual behavior | Day trips, sightseeing, rest, tourist activities. |
| | History and culture | Indicating the origin or customs of ancient towns, or art and things related to it |
| | Nature | Climate, weather, animals, plants, atmosphere, etc. |
| | Artificial | Human-related elements like small bridges, flowing water, houses, architecture, folk paintings, bridges |
| | Human | Family members, friends, tourists, local people |
| | Food | Dishes, cuisines, etc. |
| Sentiment | Positive | Sentiments such as good, like, delicious, happy |
| | Neutral | Descriptions such as uncrowded, few people, typical, and commercialized |
| | Negative | Sentiments like boring, not good, unfortunate, tired, disappointed, and regretful |

### 3.3. Data Analysis

The statistical analysis calculated the proportion of word frequencies for the three dimensions and sixteen categories in each data source. Based on the results of the statistical analysis, a two-sample z-test was conducted using the "statsmodels" library in Python to ascertain the existence of significant differences between UGC and DMO within each category. Subsequently, a correspondence analysis was performed on the four sources of data in order to assess the differences in the perception of the destination image of the two ancient towns across different data sources. Correspondence analysis allows multiple datasets to be visualized in a two-dimensional plot, using the spatial arrangement of data points on the plot to assess the degree of correlation and differences between variables [10]. In this study, all correspondence analysis plots were generated utilizing the SPSS 26 software.

### 4. Results

#### 4.1. Statistical Analysis

The results of the statistical analysis (Table 4) indicate a significant difference in the categories between DMO and UGC data (Figure 3). Among the top three categories in both sets of UGC data were "Food", "Positive", and "Individual Behavior". In ZJL-UGC,

"Food" had the highest proportion (28.74%), while in FJ-UGC, "Positive" was the most prominent (18.06%). In the DMO data for both towns, the top two categories were "History and culture" and "Artificial". However, in FJ-DMO, "Food" (12.60%) ranked third, which was very close in proportion to the UGC data for Fengjing. In ZJL-DMO, the third-ranked thematic word was "Architectural Space" (11.72%).

**Table 4.** Proportion of 16 categories.

| Dimension | Category | Proportion of Keywords (%) | | | |
| --- | --- | --- | --- | --- | --- |
| | | FJ-UGC | ZJL-UGC | FJ-DMO | ZJL-DMO |
| Space | Architectural space | 1.75 | 1.58 | 2.29 | 11.72 |
| | Gathering space | 1.31 | 0.89 | 4.20 | 3.56 |
| | Street space | 0.65 | 1.10 | 2.67 | 2.09 |
| | Waterfront space | 0.71 | 0.34 | 2.48 | 1.26 |
| | Landmark space | 1.71 | 1.61 | 12.21 | 10.46 |
| | Subtotal | 6.12 | 5.52 | 23.85 | 29.08 |
| Activity | Time | 8.95 | 8.13 | 3.24 | 1.26 |
| | External traffic | 1.44 | 0.96 | 3.24 | 0.00 |
| | Individual behavior | 16.08 | 10.45 | 6.30 | 9.00 |
| | History and culture | 5.78 | 4.64 | 24.81 | 28.03 |
| | Nature | 9.05 | 7.01 | 0.38 | 0.42 |
| | Artificial | 4.47 | 2.79 | 14.50 | 20.50 |
| | Human | 8.85 | 10.10 | 4.58 | 2.51 |
| | Food | 13.59 | 28.74 | 12.60 | 3.35 |
| | Subtotal | 68.22 | 72.82 | 69.66 | 65.06 |
| Sentiment | Positive | 18.06 | 15.60 | 3.63 | 3.56 |
| | Neutral | 3.79 | 2.94 | 2.86 | 2.30 |
| | Negative | 3.81 | 3.12 | 0.00 | 0.00 |
| | Subtotal | 25.66 | 21.66 | 6.49 | 5.86 |
| Total | | 100.00 | 100.00 | 100.00 | 100.00 |

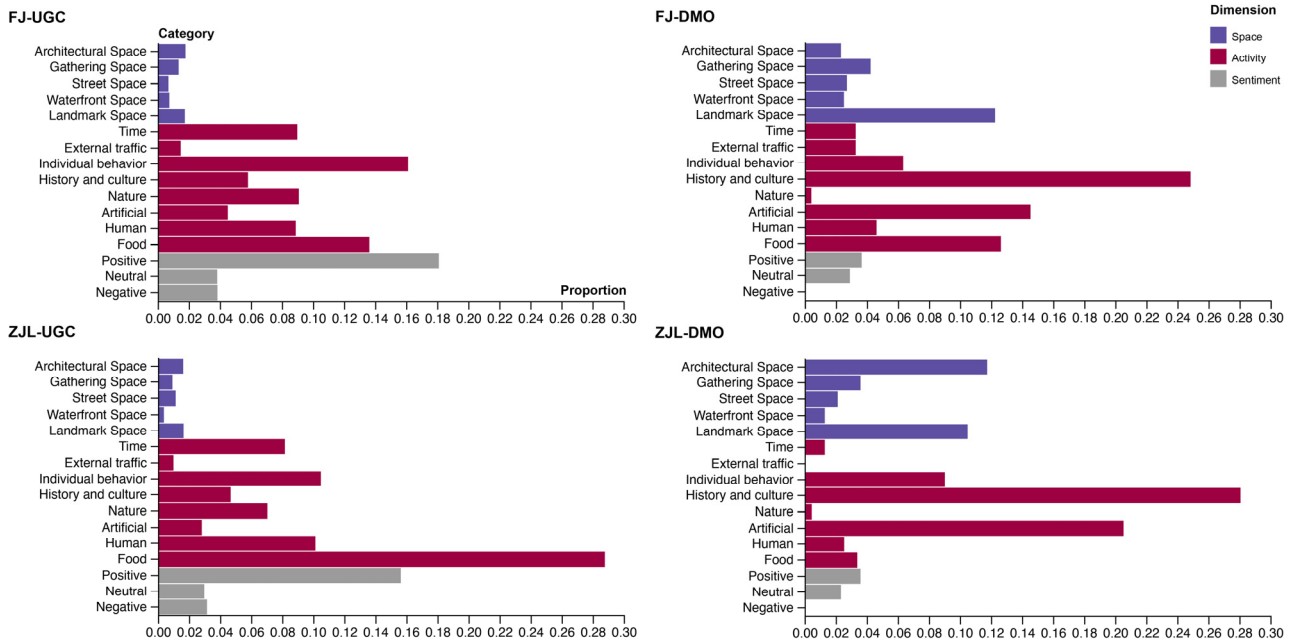

**Figure 3.** Statistical comparison chart of the proportion of keywords in 16 categories.

Based on the results of the two-sample z-test (Table 5), it is evident that in both Fengjing and Zhaojialou, there exist distinct differences between UGC and DMO. In Fengjing, all categories exhibited significant differences between UGC and DMO, except for "Architectural Space", "Food", and "Neutral". In Zhaojialou, only "Individual behavior" and "Neutral" did not display significant differences.

**Table 5.** Two-sample Z test of UGC and DMO proportion of two towns.

| Category | Fengjing | | Zhaojialou | | Alpha |
|---|---|---|---|---|---|
| | **Z-Score** | ***p*-Value** | **Z-Score** | ***p*-Value** | |
| Architectural space | −0.9157 | 0.3598 | −15.8542 | 0.0000 | 0.05 |
| Gathering space | −5.3817 | 0.0000 | −5.8204 | 0.0000 | 0.05 |
| Street space | −5.1872 | 0.0000 | −2.0148 | 0.0439 | 0.05 |
| Waterfront space | −4.4471 | 0.0000 | −3.2367 | 0.0012 | 0.05 |
| Landmark space | −15.7748 | 0.0000 | −13.9305 | 0.0000 | 0.05 |
| Time | 4.5214 | 0.0000 | 5.4801 | 0.0000 | 0.05 |
| External traffic | −3.2653 | 0.0011 | 2.1467 | 0.0318 | 0.05 |
| Individual behavior | 6.0147 | 0.0000 | 1.0235 | 0.3061 | 0.05 |
| History and culture | −16.8759 | 0.0000 | −22.2088 | 0.0000 | 0.05 |
| Nature | 6.8923 | 0.0000 | 5.6330 | 0.0000 | 0.05 |
| Artificial | −10.2577 | 0.0000 | −21.0770 | 0.0000 | 0.05 |
| Human | 3.3908 | 0.0007 | 5.4764 | 0.0000 | 0.05 |
| Food | 0.6475 | 0.5173 | 12.1717 | 0.0000 | 0.05 |
| Positive | 8.5041 | 0.0000 | 7.2117 | 0.0000 | 0.05 |
| Neutral | 1.0885 | 0.2764 | 0.8133 | 0.4160 | 0.05 |
| Negative | 4.5514 | 0.0000 | 3.9207 | 0.0001 | 0.05 |

In order to compare the relative changes in proportions of UGC and DMO data across categories, the study calculated the relative change [(DMO-UGC)/UGC] in DMO compared to UGC (Figure 4). The results aligned with the findings from the two-sample z-test, indicating that all five categories within the space dimension showed increases, with the exception of Architectural Space in Fengjing. This suggests that the DMO primarily emphasized the promotion of the unique spatial aspects of the ancient towns. However, tourists appeared to be more focused on the emotional experiences elicited by these spaces, potentially leading to an oversight of the physical spaces themselves.

For Fengjing ancient town, "Landmark space" exhibited the highest increase in DMO compared to UGC (6.162 times), followed by "History and culture" (3.293 times) and "Street space" (3.094 times). On the other hand, all categories in the sentiment dimension in DMO showed a decrease. The most significant decrease was observed in "Negative", where the frequency of the negative keywords in DMO was 0. The second-largest decrease was "Nature" (−0.958).

For Zhaojialou, "Architectural space" (6.393 times) and "Artificial" (6.359 times) showed a high increase, followed by "Landmark Space" (5.511 times) and "History and Culture" (5.044 times). Similar to Fengjing, the "Negative Emotion" category in Zhaojialou's DMO data showed one of the largest decreases, which was also because the frequency of negative keywords was 0 in DMO. Additionally, a significant decrease was observed in the "Nature" category, which was similarly notable in Fengjing. Another category that showed a substantial decrease was "External traffic", as DMO keywords did not include terms related to reaching the ancient town. Furthermore, "Food" (−0.884) is another category where DMO differed significantly from UGC.

### 4.2. Correspondence Analysis

Statistical results were analyzed according to four sources, FJ-UGC, ZJL-UGC, FJ-DMO, and ZJL-DMO, as well as three dimensions: space, activity, and sentiment (Figure 5). The results aligned closely with statistical analysis. A stark contrast between UGC and DMO in both towns was evident, indicating that the perceived tourist destination image in

the two towns differed significantly from the image portrayed by DMO. Notably, DMOs emphasized the space dimension, particularly in ZJL-DMO, reflecting the emphasis on spaces in promoting Zhaojialou. On the other hand, UGC data leaned more towards the sentiment dimension, particularly in FJ-UGC, indicating the rich emotional experiences visitors had in Fengjing. While the four sources did not exhibit significant differences in the activity dimension, it is worth noting that categories in activity dimensions dominated across all datasets. The following sections will provide a more detailed analysis of the specific categories within three dimensions.

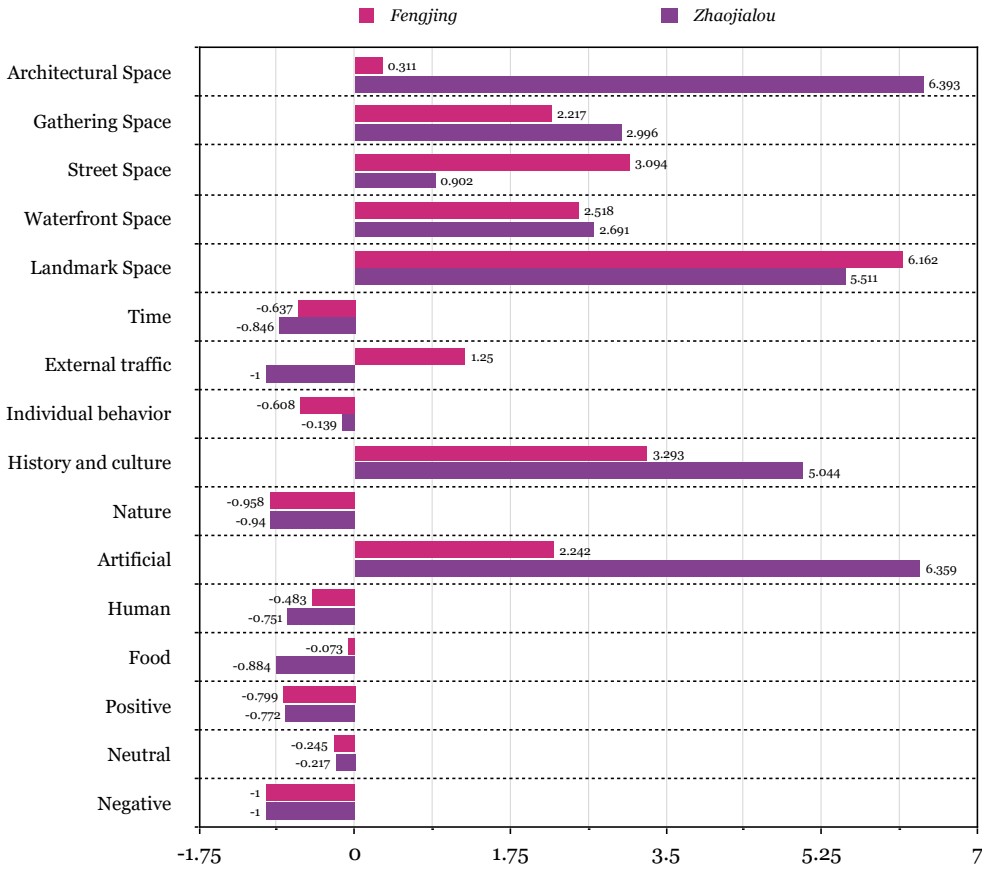

**Figure 4.** Relative change [(DMO − UGC)/UGC] in DMO compared to UGC.

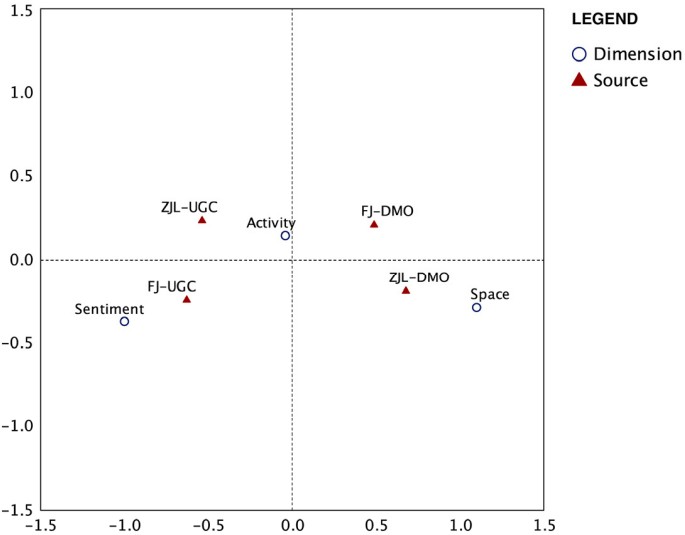

**Figure 5.** Correspondence analysis map of three dimensions.

### 4.2.1. Comparing the Space Dimension and Its Categories

The statistical results indicate that in the space dimension, DMO, in comparison to UGC, exhibited a significant increase in the proportion of categories. This increase was obvious in "Landmark space". For instance, in Fengjing, landmarks like Zhihe Bridge, Nandajie Street, and Heping Street are highlighted, while in Zhaojialou, Mei Garden, Li Garden, and Li Geng Hall are emphasized. The "Landmark Space" is positioned closer to the two DMOs in the correspondence analysis (Figures 6 and 7).

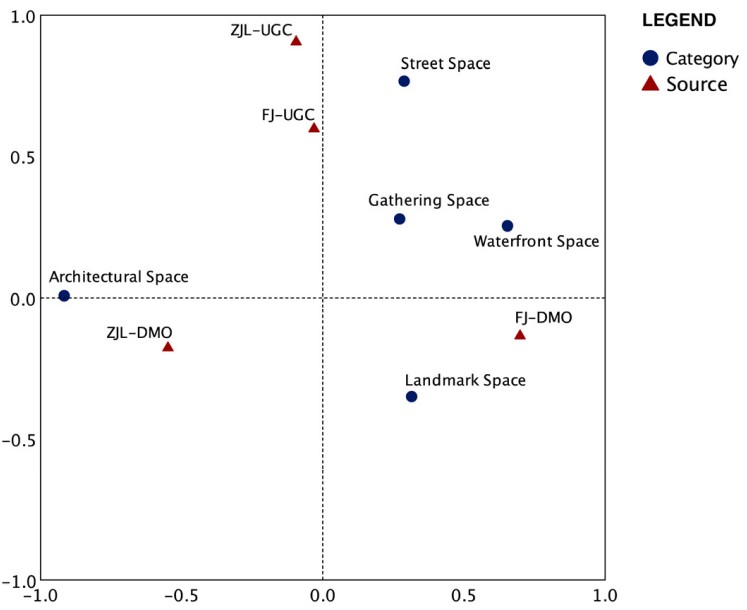

**Figure 6.** Correspondence analysis map of the space dimension.

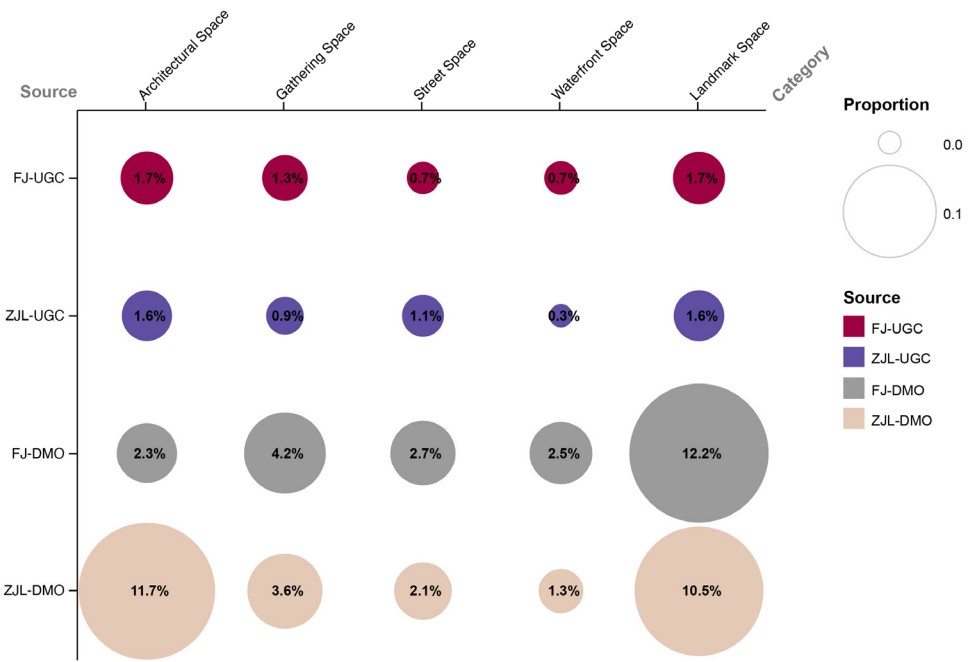

**Figure 7.** Categories in the space dimension.

ZJL-DMO emphasized spaces such as courtyards, halls, and side rooms, with "Architectural space" accounting for the highest proportion (11.72%) in ZJL-DMO, far more than ZJL-UGC (1.58%). The correspondence analysis map also demonstrates that "Architectural

space" aligned with ZJL-DMO. On the other hand, "Landmark space" leaned more towards FJ-DMO, while "Street space" was closer to both sets of UGC. No significant differences were observed in the other two spatial categories, "Gathering Space" and "Waterfront Space".

The keywords of categories in space dimension from the four sources were further compared and significant differences were found in the "Landmark space" keywords between DMO and UGC for both towns. Unlike FJ-DMO, FJ-UGC mentioned more landmarks, including "Old Stage", "People's Commune", "Corridor", "Starbucks", and "Three Bridges Courtyard". Interestingly, keywords that reflect the conjunction of Wu and Yue's cultural geographic features in Fengjing, such as boundary stones and boundary rivers, were rarely mentioned in both FJ-DMO and FJ-UGC. This suggests a lack of emphasis on these important features in both promotional materials and tourist perceptions.

The most mentioned landmarks in ZJL-UGC were dining places like "Wu Youxian" "Lao Bayang Restaurant", and "Qingju Noodle", while the most mentioned landmarks in ZJL-DMO were attractions like "Li Garden", "Mei Garden", and "Li Geng Hall". Zhaojialou also exhibited differences in "Architectural space". ZJL-DMO focused more on architectural terms like "courtyards", "alcove", "depth", and "halls", which differed from the emphasis on dining establishments in ZJL-UGC. Although other types of spaces showed varying proportions in DMO and UGC data for both towns, the content of high-frequency keywords remained largely consistent.

### 4.2.2. Comparing the Activity Dimension and Its Categories

K-means clustering was utilized to categorize keywords related to tourist activities in the ancient towns into eight categories. These categories encompass "External Traffic", "Activity Time", and "Individual Behavior", along with five categories representing aspects perceived by tourists through their activities: "Nature", "Artificial", "Human", "History and culture", and "Food". While the proportions of these eight categories varied in UGC and DMO data, the corresponding analysis map (Figures 8 and 9) clearly illustrates that the two sets of UGC and DMO data sources were distinct and situated on opposite sides of axis Y, which indicates significant disparities between the activities tourists perceived and the official promotional activities depicted by DMO.

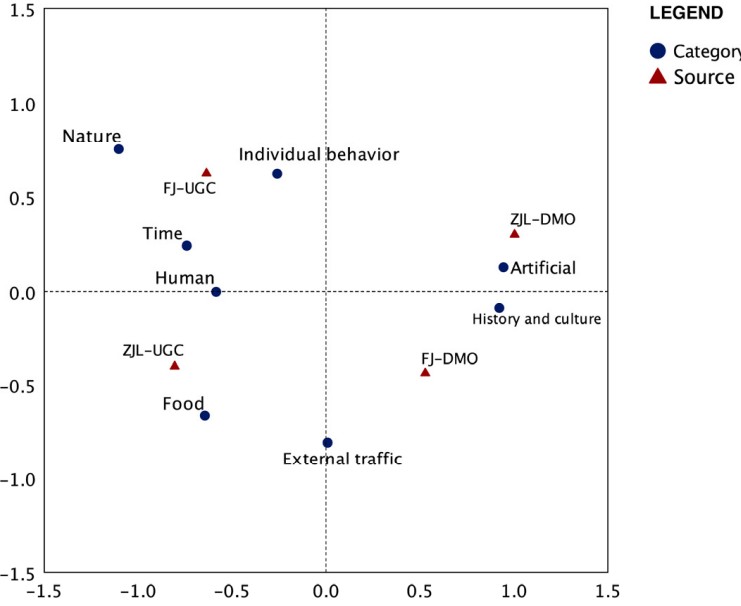

**Figure 8.** Correspondence analysis map of the activity dimension.

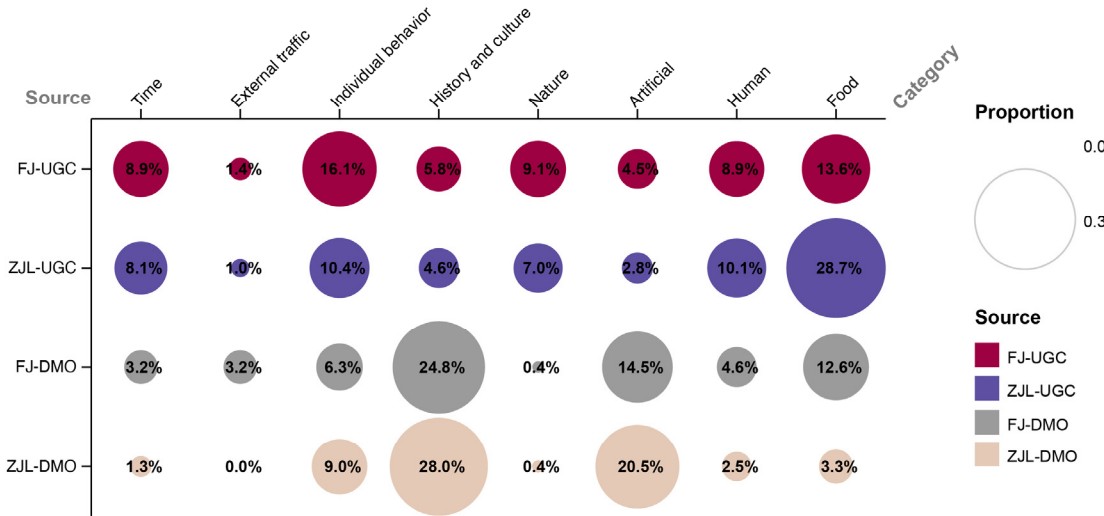

**Figure 9.** Categories of the activity dimension.

According to the corresponding analysis map, "History and culture" and "Artificial" were concentrated on the side of DMO sources, particularly in close proximity to ZJL-DMO. This reflects that DMOs, especially ZJL-DMO, perceive "Artificial" and "History and culture" as aspects that can showcase the developmental history, cultural essence, and material heritage of the ancient town, thereby highlighting its distinct features. However, in comparison to historical and cultural aspects, tourists were more focused on tourist behavior, food experiences, natural scenery, characters, and the timing of these activities (Figure 8). DMO barely mentioned elements like climate, atmosphere, plants, and animals. FJ-UGC exhibited a more noticeable focus on "Individual behavior", for example, rest, photography, excursions, and boat trips, as well as "Time", like specific days of a week or holidays. Additionally, FJ-UGC emphasized natural elements, including climate, atmosphere, and plants and animals. ZJL-UGC, on the other hand, prominently highlighted tourists' keen interest in "Food", in which keywords encompassed everyday items like Shanghai soup buns, as well as emerging trendy dishes. The distinctive traditional cuisine of Zhaojialou, known as the "Three Treasures", was also mentioned, albeit with less emphasis compared to non-traditional foods. In terms of "Human", the UGC of both towns primarily focused on close relationships, like family members and friends, with less focus on local residents or other tourists.

While FJ-UGC showed a weakness in the perception of "History and culture", the study uncovered keywords such as "love", "Nothing but thirty", and "reading" within it. These keywords correspond to Fengjing Ancient Town's distinct features, such as its Jiangnan wedding customs, role as a filming location for TV shows, and its historical significance as a hub for scholars. This underscores Fengjing's unique qualities, setting it apart from the typical "small bridges and flowing waters" image of other canal towns, which is needed for a further exploration and increased awareness among tourists. Furthermore, the widespread dissemination of movies and television, innovative wedding customs events, and the establishment of new attractions like "The Archway of Champion" have significantly contributed to enhancing perceptions of and emotional connections to the culture of Fengjing.

### 4.2.3. Comparing the Sentiment Dimension and Its Categories

A significant disparity between UGC and DMO in terms of the proportion of three categories of sentiment dimension across the two ancient towns was shown by the statistical analysis. The proportion of the sentiment dimension in UGC was much higher than in DMO: 25.66% and 21.66% in FJ-UGC and ZJL-UGC and 6.49% and 5.86% in FJ-DMO and ZJL-DMO (Table 4). "Positive" had a significant presence in both sets of UGC data,

which indicates that tourists are highly satisfied with their experiences in the two ancient towns (Figure 10), particularly in FJ-UGC, where positive emotions (18.06%) counted as the largest among 16 categories (Figure 11). Proportions of "Neutral" were similar in both towns and the proportion of "Negative" was minimal. DMO data contained no keywords related to "Negative", while the proportions of "Positive" and "Neutral" were marginally different, which suggests that DMO narratives tend to be more objective in style and may lack emotional resonance with tourists.

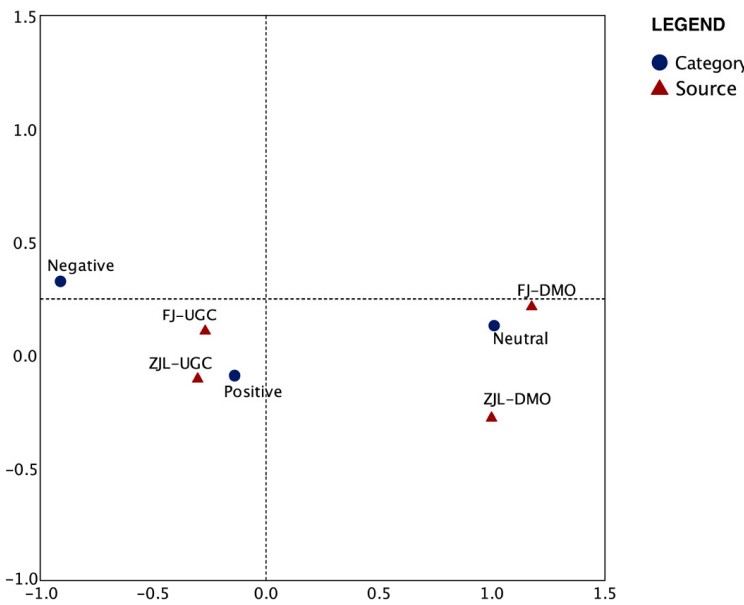

**Figure 10.** Correspondence analysis map of the sentiment dimension.

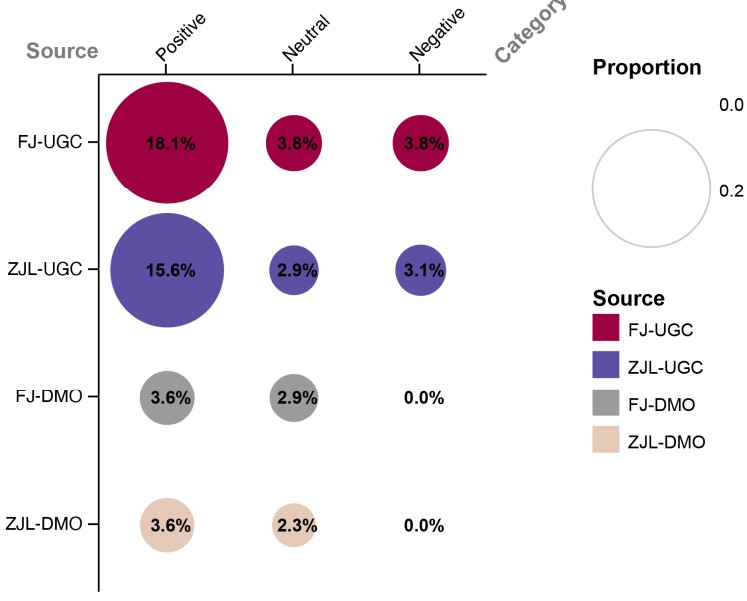

**Figure 11.** Categories of the sentiment dimension.

Among the three categories in both UGC datasets, the most significant difference lay in "Neutral". Fengjing gives tourists a sense of "quiet" and "casual", whereas Zhaojialou showcases a more bustling and commercialized tourism atmosphere with references to "crowd" and "commercialization". In terms of positive emotions, high-frequency keywords in both towns include phrases like "delicious", "happy", and "like". Perceptions of

Fengjing capture the feeling of "quaint", "tranquil", and "comfortable" that evoke senses of "pleasure", "leisure", and "happiness". In contrast, Zhaojialou brings forth "Positive" senses of "liveliness" and "enjoyment". The keyword "delicious" is highlighted 392 times, ranking first in "Positive", which echoes the outstanding presence of "Food" in the activity dimension of Zhaojialou. "Negative" perceptions of both towns share similar keywords such as "boring", "bad", "disappointing", and "regret", indicating that the tourist experience fell short of expectations. Additionally, some visitors to Fengjing felt "exhausted", while in Zhaojialou, some perceived it as "human gridlock" and "overcrowded".

## 5. Discussion

### 5.1. The Visitor-Perceived (UGC) Destination Image of Ancient Town Differs Significantly from That Officialy Promoted (DMO)

Just like previous studies have concluded, tourists focus more on sentiment [22]. Additionally, historical and cultural elements are barely shown on the list of tourists' perceptions [26], which is contrary to what DMO portrays—detailed origin, culture, and stories of the towns in a historical context. The results explain that tourists do not concentrate on historical, traditional, or cultural essence but on personal experience [40]. Due to their "immediate thought" or "immediate observation", promoted by social media, the authenticity of ancient towns is forgotten [41]. The destination image shaped by DMO can establish a brand [42]. A positive and strong local brand can create cultural value and evoke emotional attachment, thereby further boosting the sales of specific local products [43–45]. The water town Zhouzhuang, for instance, has successfully cultivated the brand of being the "Top Water Town in China", with its image of "bridges, rivers, and residential homes" being widely recognized and propagated [46]. In such a scenario, other water towns would find it challenging to compete with Zhouzhuang solely by using the spatial image of "bridges, rivers, and residential homes" to attract tourists. Fengjing aims to establish its brand as the "Millennium Conjunction of Wu and Yue", which holds significant cultural significance. However, its DMO seems to prioritize highlighting "Landmark space" while overlooking this crucial cultural symbol. Similarly, Zhaojialou Ancient Town positions itself as the "Origin of Shanghai's Cultivation Culture", but its DMO description consists mainly of specialized architectural keywords, possibly due to emphasizing its restoration project that began in 2008. It is important to note that not all tourists are professionals, from what was reported in UGC; their purpose for visiting ancient towns is primarily experiential since the large proportion of their comments did not reach the core part of ancient towns, which means the historical and cultural factors. DMO failed to resonate with tourists because their emphasis on historical and cultural aspects was not integrated into the experience of tourists, which further led to the gap between DMO and UGC. However, it is worth acknowledging that while UGC reflects immediate perceptions, they can be fragmented. In order to obtain a comprehensive understanding, these perceptions require further research and refinement.

### 5.2. Activities Evoke Sentiments Which in Turn Shape the Differentiated Perception of the Destination Image

The study revealed that the perception of the destination image of ancient towns follows a systematic process of "Space-Activity-Sentiment". As tourists engage in deeper activities, the frequency of sentiment keywords corresponding to more specific spaces gradually increases (Figure 12). The deeper their engagement in activities, the stronger their sentiment becomes. Similarly, the higher the alignment between space and activities, the more the tourists' experiences meet their expectations, leading to a stronger positive emotional response and a heightened perception of the space. Conversely, low alignment makes emotions lean toward the negative side, and the perception of the space could be diminished. For instance, consider the "Wu Yue" featured landmark in Fengjing, represented by the boundary monument and river. Since there are no spaces for visitors to stay and no corresponding activities provided in the vicinity, tourists often pass by quickly without forming emotional connections. As a result, the spaces have low perception.

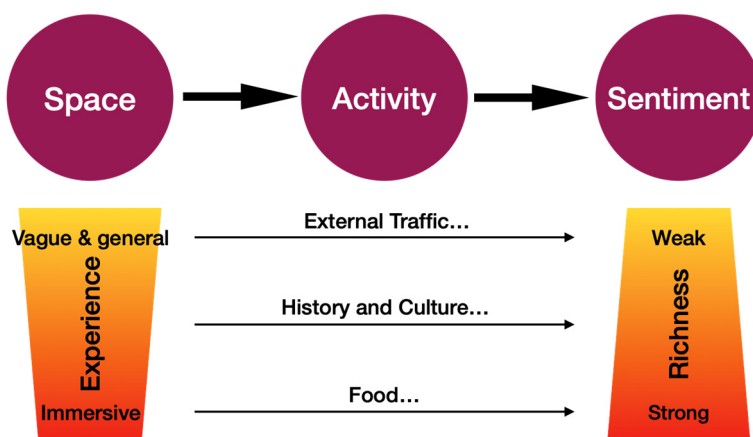

**Figure 12.** Relationship between three dimensions.

The perspective of the DMO of Zhaojialou sees culinary experiences as a small fraction (3.35%) of the town's unique offerings. However, in the tourists' perception, Zhaojialou Ancient Town is recognized as a destination famous for food (28.75%), which overlaps with the previous research [27]. This branding image does not have a direct connection with the town's historical and architectural attributes.

*5.3. The Culture of the Ancient Town Continues and Regenerates through Tourism*

The differences between DMO and UGC reveal that some non-traditional and emerging elements have captured greater interest among tourists. For instance, in Fengjing, tourists mentioned "Nothing but Thirty", Starbucks, "Three Bridges Courtyard", and "People's Commune". Additionally, "Wu You Xian", "Lao Ba Yang Restaurant", soup duns, and shumai were mentioned by tourists in Zhaojialou. These elements are not the traditional aspects of these ancient towns, yet they are prosperous. Media plays a crucial role in establishing and maintaining the "intense pleasure" that people associate with a destination, often through means such as movies, TV shows, or travelogs [47]. Fengjing Ancient Town, being a location featured in various forms of literature and art, has been created and given new cultural value alongside these works. This has sparked emotional resonance among audiences. When these audiences visit Fengjing in person, the scenes they encountered in media will rekindle their emotional attachment to the place. Similarly, the People's Commune Memorial Museum in Fengjing has the same emotionally evocative effect on visitors who have experienced the historical phase of the people's commune. Tourists are drawn to Zhaojialou to experience emerging cuisines famous on social media. Food production is inherently a part of the cultural landscape, and dining experiences during the tourism process can fulfill not only material needs but also cultural and social desires of visitors [48–50]. The initial formation of ancient towns was primarily for the exchange of goods among the surrounding rural residents, and temporary markets gradually evolved into permanent shops, eventually developing into commerce-centered communities [51]. Ancient towns provide a physical space where people engage in "exchange", leading to the development of their culture. A theory from heritage tourism has also claimed that experience needs to be managed, which means that heritage tourism has the potential to revitalize a destination, enhance its infrastructure and environment, and serve as both a revenue generator and a source of employment, and in this way, historically significant buildings, existing craft expertise and traditional performances will not disappear for lack of social, political and financial support [40,52,53]. In the cases of Fengjing and Zhaojialou, new elements suit the preferences of contemporary tourists and encourage them to come and "exchange". During this process, the cultural essence of the ancient towns is sustained, and new cultural elements may emerge as a result.

## 6. Conclusions

Theoretically, the results confirm the former research findings that a gap between tourists' perception (UGC) and the official promotion by Destination Marketing Organization (DMO) exists [10,54]. Tourists often share content in their travel log that deviates from the image crafted by destination marketing efforts. DMOs should adjust their promotional strategies based on tourist perceptions and the actual situation. They should go beyond focusing solely on presenting general but vague descriptions of ancient towns, instead emphasizing the incorporation of emotionally rich content, contributing to the development of a stronger destination image [10,55]. Lastly, engaging in appropriate interactions with UGC platforms fosters emotional connection and identification with tourists, thereby effectively guiding the formation of the destination image. Furthermore, while "Negative" offers direct and accurate insights into tourist preferences which can aid DMOs in adjusting planning strategies or policies [20], it is important to recognize that emotional expressions on online platforms can be overestimated and inclined to criticize destinations with negative emotions. DMOs should interpret the perceptions and preferences of tourists reflected in UGC content objectively and comprehensively.

Practically, the breakthrough of this study lies in its utilization of a machine learning approach to analyzing online data, especially for the ancient towns. The acquisition of big data frees the research on destination image from the constraints of traditional small-scale surveys, granting this kind of research increasing credibility and objectivity. The data-driven classification approach employed in this study effectively combines the content of comments, aligning with the principles of grounded theory [10]. For handling larger volumes of data, employing machine learning approaches tends to yield more comprehensive and objective results. The analysis of UGC also provides novel insights for the managers of Fengjing and Zhaojialou, motivating them to explore tailored strategies for improving the tourist experience in these historic towns. Moreover, it aids in the identification of previously overlooked areas, enabling the integration of their planning strategies with tourist preferences in future endeavors. This, in turn, promotes the balanced development and preservation of these ancient towns, ultimately offering valuable insights for the protection and development of other similar historic sites.

There are certain limitations in this study, such as that the process of extracting and analyzing keyword content could potentially overlook nuances within the context and language used. Moreover, interventions based on expert knowledge and experience might be necessary to accurately classify keywords. Additionally, given the relatively narrow scope of UGC content sources, this study might inadvertently exclude insights from tourists beyond those using Sina Weibo. Future research will strive to encompass a broader range of UGC platforms, encompassing diverse content forms such as travel logs and photographs, thereby capturing a more multifaceted panorama of tourist perceptions. Furthermore, in order to let the machine learning fit more on specific tasks, pre-trained language models which provide fine-tuning functions should be considered in the future.

**Author Contributions:** Conceptualization, J.D. and L.W.; methodology, J.D.; software, J.D.; validation, J.D., L.W., Z.T. and D.C.; formal analysis, J.D., M.H. and D.C. investigation, J.D. and L.W.; resources, L.W.; data curation, J.D., D.C. and Z.T.; writing—original draft preparation, J.D.; writing—review and editing, J.D. and L.W.; visualization, J.D.; supervision, L.W.; project administration, L.W.; funding acquisition, L.W. All authors have read and agreed to the published version of the manuscript.

**Funding:** This research was funded by the Shanghai Planning Office of Philosophy and Social Science, grant number 2019BCK001.

**Data Availability Statement:** Not applicable.

**Conflicts of Interest:** The authors declare no conflict of interest.

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
