# Peer review of "A Comparative Study of Perceptions of Destination Image Based on Content Mining: Fengjing Ancient Town and Zhaojialou Ancient Town as Examples"

_land, doi:10.3390/land12101954_

Round 1

Reviewer 1 Report

While text analysis of DMO and UGC content is nothing new, the reviewer appreciates the effort taken by the authors to consider two ancient towns in Shanghai. 

Since the analysis was done in Chinese language, but the readers of the article may not be Chinese speakers, I would find it interesting to highlight the differences in text analysis (e.g. the word embeddings) when done with Chinese text compared to English language. At the very least, be more explicit that you collected Chinese texts (Mandarin, to be precise?). Since Chinese uses Hanzi characters rather than an alphabet, does this affect text analysis approaches? Is there more ambiguity in the meaning? I assume also you used a word2vec pretrained with Chinese text, but you do not state it clearly. For the international reader, it is interesting to know if there were any (Chinese language specific) issues in the text analysis stage. 

Table 3. In English we want to avoid "gender biased" phrases. So "human made elements" and not "man related".

Size of the DMO data is much smaller than the UGC data it is compared to, statistically this is limiting when aiming for a comparison. You didn't state anywhere number of characters in each data set. But if DMO was 4-5 articles and UGC is text over 10 years, the scale of the datasets must be very different. This needs to be explicitly addressed when doing statistical comparisons. 

Fig 3. If you put DMO and UGC values side by side, it is easier to compare.

Fig 4. The statistical approach is rather naive. To compare two proportions to say if they vary significantly, use a statistical test such as the two sample z test. If you assume the distibution of the categories in the text would randomly tend to be equal (but this is an assumption), you could use a chi squared test.

Correspondence analysis is a relevant technique to visualise and interpret the data. Please add the software or code library you used to create the maps, together with any parameters of relevance (remember science is built on the idea that others can replicate your work)

In conclusion, I think this work needs to address some issues. If the DMO datasets are much smaller than the UGC, I would explicitly address this in some way - at the least, your statistical tests comparing both datasets must take the difference in size into account. 

Very well written with good use of English language. Only very minor errors which might be fixed with a good proof-reading tool, e.g. "China mainland" --> we tend to say "mainland China". However nothing affects the understandability of the article. 

Reviewer 2 Report

This manuscript holds significant potential for addressing a crucial theoretical gap. However, specific refinements can enhance its overall quality.

The introduction should provide a more comprehensive exploration of this theoretical gap, despite the numerous innovations presented.

In the literature review, the emphasis should pivot towards the subject matter rather than an excessive focus on methodology. Drawing from noteworthy studies in the field can enrich the overall discourse.

To improve the manuscript, consider allocating more depth and focus to the discussion section. Strengthening your argument with a solid foundation in the literature can refine the quality of this section.

I recommend placing greater emphasis on the manuscript's theoretical aspects, thereby making your theoretical contributions more distinct.

Reviewer 3 Report

The topic of this article is interesting and very current. Practical implications are valuable because they indicate the differences between the image propagated by promotion and the one that has been formed in the minds of tourists. The technique used can be an inspiration for other researchers regardless of the region and the nature of the place - of course, after some modifications.

The article is correct in terms of methodology, content and language.

There are a few minor shortcomings and wordings that are unclear/imprecise in the text. After their modification/correction, the article may be published.

Reviewer 4 Report

land-2649465-peer-review-v1

Review of: A Comparative Study of Perception of Destination Image 2 Based on Content Mining

This is a nice study that exemplifies the differences between marketing and actual experiences. It provides a good case study that could be expanded to other neighbouring historic communities. Subject to the language comment I have no concerns and would be keen to see this in print.

The paper needs a thorough edit by a scientific, native English speaking editor (not just a native English speaker). It is riddled with infelicities in grammar and expression that can be easily fixed.

The paper needs a thorough edit by a scientific, native English speaking editor (not just a native English speaker). It is riddled with infelicities in grammar and expression that can be easily fixed.

Reviewer 5 Report

Dear Author(s),

Thank you for the opportunity to read the paper entitled A Comparative Study of Perception of Destination Image Based on Content Mining: Fengjing Ancient Town and  Zhaojialou Ancient Town as Example. I found this paper very interesting. The topic of this paper is interesting, but certain improvements would be appreciated.

Abstract

Comment 1

The abstract gives impression of an interesting article, but it needs to be shortened. Try to keep it within 250 words.

Introduction

Comment 2

The Introduction needs to present the background of the studies in a similar context. The research gap needs to be highlighted using examples that are context-based.

You should point out the contribution, novelty and originality of the study.

Methods

Comment 3

Research design is appropriate. The assessment framework is detailed and explained. The figure is very nicely presented.

Results

Comment 4

Figure 3 – Can you adjust the letters and make them bigger? It is hard to read.

Comment 5

The results are nicely presented through tables and figures. Great job!

Conclusion

Comment 6

The implication section needs to be improved. What are contributions to theory? Practical implications?

Comment 7

Once again, thank you very much for the opportunity to read this interesting article. Looking forward to reading your article again.

Wish you all the best!

Sincerely,

Reviewer 

Round 2

Reviewer 1 Report

Thank you for your response to reviewers' comments. My comments have been resolved. 

line 184: "All of the text materials are consisted by Mandarin Chinese characters." should be "All of the text materials consisted of Mandarin Chinese characters."

line 530: "is existed" --> "exists"

Author Response

Dear Reviewer:

Thank you so much for taking the time to review this manuscript again. We are truly grateful for your valuable suggestions. Your insightful feedback has been instrumental in improving the quality of our article. We extend our heartfelt gratitude for your patient and meticulous assistance during the revision process. In accordance with your latest comments, we have made the necessary modifications to our manuscript.

Point-by-point response to Comments and Suggestions for Authors

  • line 184: "All of the text materials are consisted by Mandarin Chinese characters." should be "All of the text materials consisted of Mandarin Chinese characters."

Thank you for pointing out the error, we have revised the sentence.

  • line 530: "is existed" --> "exists"

Thank you for pointing out the error, we have revised the sentence.

Once again, thank you for your thoughtful review and invaluable recommendations.

Warm regards,

Authors

Reviewer 2 Report

Dear Editor

I hope this message finds you well. I wanted to express my thoughts and provide a recommendation regarding the manuscript titled "A Comparative Study of Perception of Destination Image Based on Content Mining: Fengjing Ancient Town and Zhaojialou Ancient Town as Example" submitted to Land. I had the opportunity to reevaluate the manuscript and observed significant improvements made by the authors regarding quality and content.

Given the substantial enhancements, I recommend accepting this manuscript for publication on land. The authors have addressed previous concerns effectively and have elevated the overall quality of the paper. I believe this work would make a valuable contribution to the journal and benefit the scholarly community.

Thank you for considering my recommendation, and I look forward to seeing this valuable research published in the land.

Sincerely,

Author Response

Dear Reviewer:

We sincerely appreciate your message. Thank you for dedicating your time to reevaluate the manuscript. Your insightful feedback has played a pivotal role in enhancing the quality of our article. We would like to extend our heartfelt gratitude for your patient and meticulous assistance in the revision process.

Warm regards,

Authors